# Service Users' Perspectives on the Implementation of a Psychoeducation Group for People on the Waiting List of a Specialist ADHD Service: A Pilot Study

Bethany Gore [1], Frederick Omoni [1], Jemma Babiker [2] and Jon Painter [1,*]

[1] Department of Nursing and Midwifery, Sheffield Hallam University, Sheffield S11 9BF, UK; bethany.l.gore@student.shu.ac.uk (B.G.)

[2] ADHD Service, Sheffield Health and Social Care Trust, Sheffield S11 9BF, UK

* Correspondence: j.painter@shu.ac.uk

**Abstract:** In the UK, Attention Deficit Hyperactivity Disorder and waits for assessment, diagnosis, and treatment are all growing problems. This study set out to gather service users' suggestions as to how one specialist ADHD service could improve the experiences of people on their waiting list. Following a semi-structured focus group, an inductive thematic analysis of data yielded three themes: (1) support for psychoeducation in principle, (2) psychoeducation regarding the wider, holistic impact of ADHD, and (3) suggested structures and approaches, as well as (4) a set of general feedback that could inform service developments. Service users supported the notion of psychoeducation sessions to inform people on the waiting list about the wide range of potential impacts of ADHD, the most common psychiatric comorbidities, some potential coping strategies they could try, and the service they could ultimately expect to receive. Some form of one-to-one telephone support was also advocated, primarily to address their concerns about the lack of individualisation group psychoeducation could offer. The potential benefits of these suggestions combined with the low risk of adverse effects makes group psychoeducation a worthwhile waiting list initiative. However, as with any service development, it should be piloted and evaluated before being termed treatment as usual for the service.

**Keywords:** adult; ADHD; psychoeducation; service evaluation; waiting lists; group work

## 1. Introduction

Attention deficit hyperactivity disorder (ADHD) is an early onset neurodevelopmental disorder that persists into adulthood in about 50% of cases [1]. Estimates of its global prevalence range from 2–7% [2]. Clinical guidelines advocate the integration of psychosocial and pharmacological treatments [3].

Three somewhat overlapping systematic reviews into non-pharmacological interventions for adult ADHD were published in 2020 by Nimmo-Smith et al. [4] Fullen et al. [5] and Scholz et al. [6]. According to Nimmo-Smith et al. (2020), the literature is dominated by studies of cognitive behavioural therapy, though Fullen et al. (2020) observe that even these lags behind the evidence base for its use in other conditions [4].

Psychoeducation is the provision of information regarding a patient's disorder, its cause and related dysfunctions and deficits [6]. It is integral to many of these non-pharmacological interventions [5]; however, it is also considered to be an intervention in its own right which, despite a relative dearth of evidence for its effectiveness, is increasingly being seen as a first line intervention [6]. Collectively, these three systematic reviews identified just five suitably robust studies into psychoeducation one of which only included it as part of a broader intervention, i.e., Salomone et al. [7].

Hirvikoski et al.'s RCT found psychoeducation to be more effective than 'treatment as usual' in increasing knowledge about ADHD and global life satisfaction [8]. More than 90%

of their participants completed the course and reported high levels of satisfaction with the intervention. However, only people who could attend with a 'significant other' were able to participate. Vidal et al. compared psychoeducation to group CBT [9]. They found both interventions produced a significant improvement in the core symptoms of ADHD, and its associated co-morbidities. Effect sizes were, however, small, and the study was described as a pilot. The remaining two randomised controlled trials both compared psychoeducation to mindfulness. Hoxhaj et al. found the two interventions to be equally effective in reducing core ADHD symptoms with women benefitting more than men [10]. Similarly, Bachmann et al. found both interventions to be equally effective in improving working memory function and decreasing core symptomology [11]. All five studies, however, excluded individuals without a formal diagnosis, meaning they have limited applicability for people waiting for diagnosis and all advocated more research into the intervention.

What is even less clear from a brief search of the literature is precisely what service users feel should be covered in an adult ADHD psychoeducation package and what benefits they expect to derive from attending. In the UK, specialist adult ADHD services are under considerable pressure with lengthy and rising waiting lists [12]. This has led a local ADHD service to consider what they can offer to service users prior to their diagnostic appointments. In the absence of any additional funding, the aim of this pilot study was to ascertain what a group of current service users felt was lacking, in this regard, from the local adult ADHD service they had received.

## 2. Materials and Methods

This MSc dissertation project employed a qualitative service evaluation design, so that an understanding of the subjective experience of participants could be gained [13,14]. Ethical approval was granted by Sheffield Hallam University (UREC 3) and the project was registered as a service evaluation by SHSC Trust (06122021).

### 2.1. Recruitment

Purposive sampling was utilised as only those with a diagnosis of ADHD within the Sheffield ADHD service, that had capacity, and were English-speaking were contacted [15]. Those that were waiting to be assessed were excluded as they would not have sufficient insight into the service in question.

Potential participants were initially contacted by post and/or telephone between Dec 2021 and February 2022 to inform them about the study. Once sufficient interest had been expressed to form a meaningful focus group [16], copies of the participant information sheet (PIS) and consent forms were sent via post. As per the Mental Capacity Act (2005), capacity to consent was assumed unless prior contact with the service had led staff to express legitimate concern.

### 2.2. Data Collection

Due to time and resource constraints, a single online semi-structured focus group consisting of four males and one female was held by the researcher (who was independent of the service). Mindful of the challenges that group work might pose to the participants with ADHD, the focus group began with a briefing that included ground rules and the expectation that everyone would be supported to both express their views, but also listen to those of others. The Zoom platform [17] was also used to record and transcribe the data, with a separate audio recorder as a back-up. Following introductions and a discussion about ethical issues (such as confidentiality, anonymity, and participants' right to withdraw); the following questions were asked, as informed by the literature and the research question:

1. What do you know about psychoeducation as an intervention?
2. How do you think you would benefit from being educated more thoroughly on your condition?
3. What topics do you think should be included within a psychoeducation intervention?

4.  Is there anything else that anyone would like to discuss regarding the ADHD service or psychoeducation?

Supplementary questions were asked by the researcher as appropriate, with the focus group lasting a total of 46 min after which a short debrief was conducted. A nurse consultant from the service (who had previously reviewed the questions) was present for the brief and debrief but not the main discussion to encourage candour. They were, however, constantly available in a separate Zoom break-out room in case of participant distress.

*2.3. Data Analysis*

The Zoom transcription was reviewed and corrected using the audio recording to produce a verbatim account of the discussion in Microsoft Word [18]. An inductive thematic analysis was then performed using Braun and Clarke's six-phase method [19], i.e.,

(1) Familiarising self with data: the transcript was read repeatedly to become immersed in the content.

(2) Generating initial codes: important statements were highlighted, and the transcript re-read to confirm all key phrases had been selected.

(3) Constructing themes: the key statements were colour coded, to represent initial sub-themes. Preliminary names were given, and the transcript reviewed to ensure all relevant statements had been included.

(4) Reviewing themes: the homogeneity of each set of codes was considered and some statements were moved between themes until they were in the theme that best described the quote. Sub-themes were also combined to create larger themes.

(5) Defining and naming themes: sub-theme and theme titles were reviewed to ensure they accurately represented the source data (codes), were sufficiently distinct from each other, and all spoke to the over-arching research question.

(6) Producing the report: salient quotes for each theme were selected to aid their description and to provide rationale for subsequent practice recommendations.

## 3. Results

Three primary themes, each with several subthemes, emerged from the thematic analysis: (1) support for psychoeducation in principle, (2) the need to cover the wider, holistic impact of ADHD, and (3) suggested structures and approaches. General feedback regarding the ADHD service was also gathered and anything related to the research question is presented as an additional theme, (4), to help inform the proposed intervention (see Figure 1).

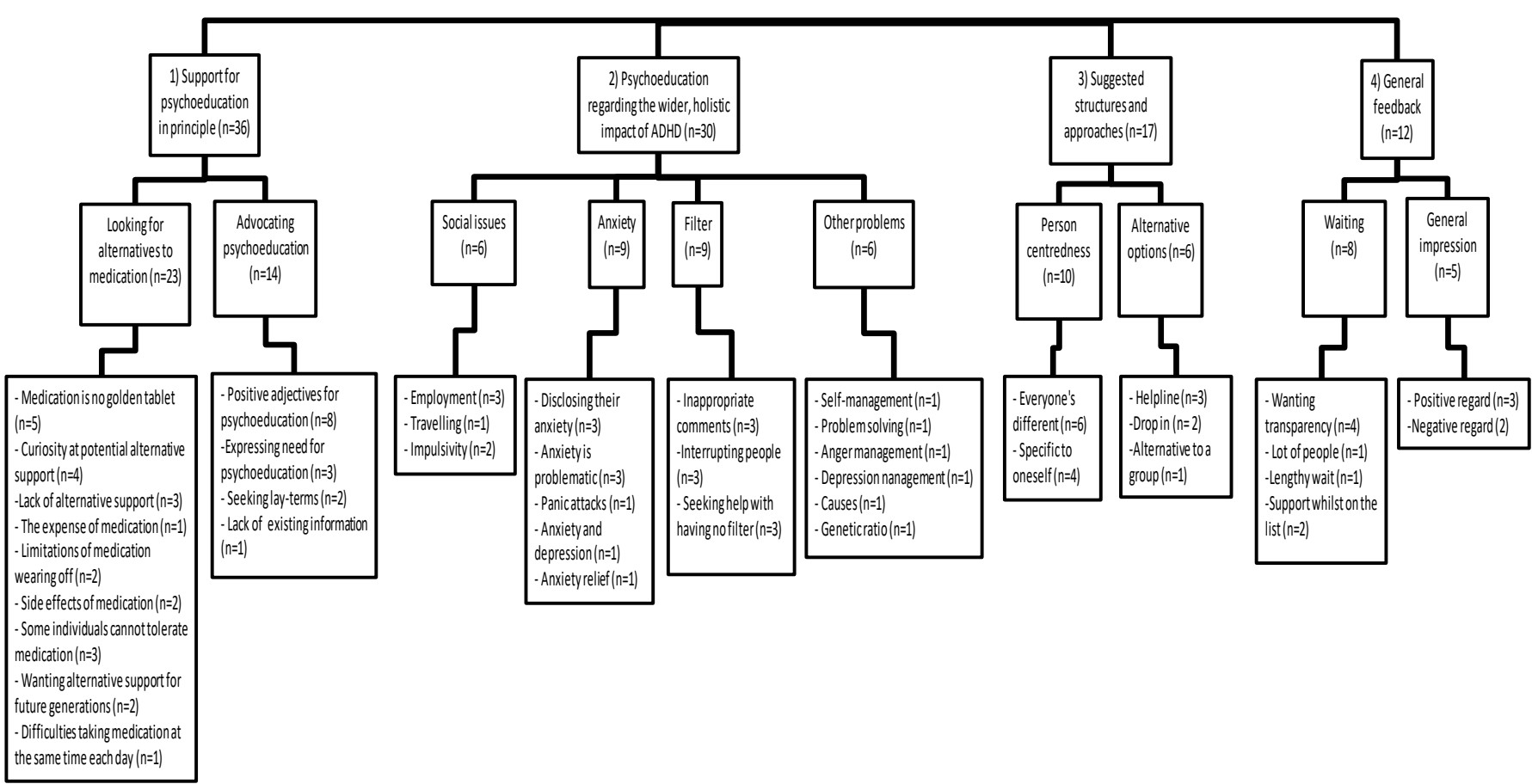

**Figure 1.** Coding tree.

### 3.1. Support for Psychoeducation in Principle

Participants acknowledged that the ADHD team was currently only commissioned to provide a diagnostic and medication initiation service. However, they had all experienced limitations and/or adverse effects of pharmacological interventions and were keen to consider psychoeducation as an augmentation strategy.

Participant 2—" ... *there's a lot of people out there that medication certainly isn't the answer. And if we can come up with something that works instead of medication, I think it's wonderful*".

Participant 3—"*I think it's just at the minute it's just about well, medication, but sometimes medication's not just the answer*".

Participant 4—"*I think it could compliment medication as well. I think there's, I think there's a need to say that there needs to be the choice to have, have them both together combined*".

Specifically, participants welcomed the potential to receive trustworthy information about their condition, perceiving this to be a means of empowerment.

Participant 4—"*I think understanding it in a broad sense to begin with is helpful. I mean, when, when I had my assessment about less than 12 months ago now, you get a lot of information that is, like medical speak. And the assessment went quite quickly in the end, or seemed to go quite quickly, so having a baseline education of it would be useful but would just be a starting point*".

Participant 1—"*For me, clearly more information will be great. Yeah, there's a lot of misinformation out there there's a lot of pseudoscience there's a lot of things that I would like to know*".

Overall, there was clear support for the provision of psychoeducation at an early stage in the service user's journey.

### 3.2. Psychoeducation Regarding the Wider Holistic Impact of ADHD

In addition to information about ADHD itself, participants stressed the need to inform users about common comorbidities.

Participant 3—"*I won't say the ADHD is the problem, it's the underlying issues like the anxiety, the paranoia, the confidence*".

Participant 2—"*I've had anxiety issues, since I was around 20, and I struggle I would say more with the anxiety, than the ADHD*".

They were also concerned that users were made aware of the diverse ways that ADHD had impacted their lives.

Participant 1—"*I have a full-time job, it's really difficult*".

Participant 2—"*I have a great difficulty travelling*".

As well as how to manage the primary symptoms of both.

Participant 1—"*I would very much like, yeah, help on the [cognitive to verbal] filter*".

Participant 2—"*Well I've gone through life, and you see it in the cartoon with the devil on one shoulder angel on the other. I'm sad to say that the devil wins every time. I've always been told all through life that I have no [cognitive to verbal] filter*".

Participant 1—"*And just more mental health support, more group therapy, more thoughts on how to do anger management, how to manage yourself, even I'd take even like basic help with diet planning sometimes*".

Participant 3—"*Whether you put action plans in, problem solving*".

Participant 5—"*About managing the depression and the hyperactive side*".

Participant 3—*"Learn how to self-manage things"*.

Together, the participants provided helpful insights into what should be included in a psychoeducation package. Additionally, that there would be benefits to users receiving this input at an early stage of their journey through ADHD services.

### 3.3. Suggested Structure and Approaches

Participants appreciated resource constraints were driving the notion of a psychoeducation group, rather than 1:1 delivery; however, there were some reservations. Primarily, these concerned the idiosyncratic way ADHD had affected participants, and the fact that a group intervention might preclude the provision of person-centred, individually tailored information.

Participant 1—*"Everyone's different so what works for one person won't work for another"*.

One participant described his positive experience of (privately funded) 1:1 psychoeducation.

Participant 4—*"Like I want something that is specific, and it's all very intense, an hour and a half session every week. However, it's very very practical advice, you know, and it's tailored to you, what my own process is like what my triggers are and what my strengths are really"*.

Which led the group to make a number of suggestions to ameliorate these shortcomings.

Participant 3—*"Maybe a drop in. Well, you know, you know that doesn't cost a lot of money does it to have a drop-in centre"*.

Participant 3—*"Somebody on the phone once a week that can answer some of these questions"*.

Participant 2—*"Again making that a part of the service where, if someone is having a bit of a crisis time with things and they really do need to speak to someone that there is actually a number that somebody will speak to you"*.

Overall, therefore, a group intervention was supported by participants, provided sufficient attention could be paid to individual attendees.

### 3.4. General Feedback That Could Inform Service Developments

As part of the focus group, participants were given the opportunity to provide general feedback about the service, all of which was passed onto staff. Presented here are just the points that speak directly to initiating a psychoeducation group for people on the ADHD service's waiting list.

Participant 4—*"I want to be able to know what I'm actually going to be waiting for, and what choices there are for me to be on those waiting lists"*.

Participant 1—*"Yeah, I mean the transparency is, it's just a blank point, basically I have no idea why I'm on the waiting list or what's going to happen or what the programmes are or where it's going"*.

Participant 3—*"But for the people that are waiting for medication. Maybe a group could be okay. Because then they're not just getting ignored"*.

Lengthy waits were a definite source of dissatisfaction but participants felt that more detailed information about the interventions, they could ultimately expect would make the wait less frustrating, hence appreciating the efforts being made with this project.

Participant 3—*"They are an amazing team, they're trying to get, this is why you know people are here today. They are trying to, you know, do the best for the service users"*.

## 4. Discussion

In the spirit of co-production [20], this study sought to understand what service users felt should be included in a psychoeducation group for individuals on the waiting list of a specialist ADHD service. Utilising a focus group, five service users were encouraged to

reflect on their experiences to evaluate the service they had received, particularly focusing on any deficits in the information they had received about their diagnosis in the early part of their care pathway. Following an inductive thematic analysis of the verbatim transcript, four themes emerged.

**Support for psychoeducation in principle:** The National Institute for Health and Care Excellence (NICE) advocates group-based ADHD-focused carer support and states this need not wait for a formal diagnosis [3] (p. 10). However, for service users themselves, it recommends information about the potential impact of ADHD should be given post-diagnosis [3] (p. 13). This contrasts somewhat with our participants who perceived definite benefits to educating individuals whilst waiting for their diagnosis, about ADHD, its common comorbidities, and the impact both can have on their lives. NICE does, however, qualify its advice with an acknowledgement that such information may still be of benefit to people with symptoms of ADHD that fall short of the diagnostic criteria [3] (p. 14). Furthermore, doubt has been cast on the validity of their previous ADHD guidance [21], much of which remains unchanged in the latest version. Cortese et al. also support our participants' views by proposing that pharmacological and non-pharmacological interventions should be considered as complimentary [22]. However, even psychoeducation in isolation has been found to improve symptoms and functioning in well-designed studies [4–6,9–11,23,24]. Set against the current backdrop of the excessive waits for specialist ADHD diagnostic services, evidence of effectiveness, and lack of iatrogenesis it therefore seems reasonable to instigate some form of structured information giving for people experiencing problems typically associated with ADHD but who are yet to be formally diagnosed.

**Psychoeducation regarding the wider, holistic impact of ADHD:** Our participants cited anxiety as a frequently encountered problem. This is contradicted by some works of literature which found no significant association between ADHD and anxiety [25]. However, it chimes with other studies that suggest it is the most common comorbidity experienced by individuals diagnosed with ADHD [26]. Unsurprisingly, individuals with concurrent anxiety have been shown to be more symptomatic, have lower occupational achievements, and increased levels of anger [27]. Help with anger management was also identified in our study, fitting with other research that found emotional dysregulation and outbursts of temper to be common in people with ADHD [28,29].

The wide-reaching effects of ADHD were universally recognised by our focus group participants which is again, in keeping with the literature which highlights deficits with executive functioning and more workplace absence, leading to reduced productivity and work performance and higher unemployment levels [30,31]. Whilst medication can improve performance at work in comparison to placebo [32], psychoeducation can aid occupational functioning [33] without the adverse effects that medication can produce. Collectively, previous research adds weight to the recommendations made by our participants regarding the breadth of content required in a psychoeducation package for people experiencing symptoms of ADHD and its most common comorbidities.

**Suggested structures and approaches:** In the absence of direct experience of group interventions, our participants had some reservations about the delivery of psychoeducation in a group setting. These pre-conceptions focused on whether group interventions could ever really be person-centred. However, Byrne et al. found control and power over one's own care to be integral to the concept [34], which Chiocchi et al. suggest can be achieved through group psychoeducation programs [35]. Furthermore, the literature identifies benefits that may only be realised through this treatment modality. There is also evidence that users can each derive different benefits from the same group intervention [36,37]. It seems, therefore, that whilst group interventions may suit some users more than others, they need not be a 'one size fits all' experience that precludes person-centredness. Participants also offered several ways to ameliorate their concerns, including augmentation of the group with some form of helpline. Elsewhere, telephone helplines have proved to be cost-effective [38] and particularly suited to supporting people waiting for mental healthcare [39], making them a viable consideration for this particular service.

**General feedback that could inform service developments:** Waiting times for mental health services in the UK are a well-recognised issue [12]. Unlike some other studies, our participants did not explicitly state their mental health had been adversely affected. Their frustration was, however, palpable [40]. Helpfully, they were able to provide insight into the experience of waiting to be diagnosed and what could make this time less anxiety-provoking for others. In addition to the topics described in theme 2, this centred around the provision of a clear explanation of the diagnostic (and medication) service they would go onto receive [41]. Service users understandably expect a prompt, efficient service [42,43]. In the current climate, waiting lists are somewhat inevitable however, it seems that transparency and high-quality information may reduce their negative impact.

*4.1. Recommendations for Practice*

This service evaluation project was specifically contrived to provide tangible options for an ADHD service to consider as part of their waiting list management plans. Based on the views of a group of their service users, and a brief review of the relevant literature our suggestions are that:

- A psycho-education group for people on the waiting list for a diagnostic appointment should be piloted.
- The content should include:
  - a detailed explanation of the service that people can ultimately expect to receive;
  - information regarding ADHD and the full breadth of its potential impact on individual's functioning and quality of life;
  - information about the most frequently encountered psychiatric comorbidities;
  - low-level coping strategy enhancement
- Consideration should be given to augmenting the group intervention with some form of one-to-one telephone support, perhaps linked to periodic updates of people's position on the waiting list.
- A further student service evaluation project should be devised to ascertain the impact of group psychoeducation for people waiting for specialist ADHD assessment and diagnosis. This should adopt a mixed methods approach to capture subjective and objective data.

Out with these service-specific recommendations, it is clear that larger scale studies with well-matched control groups are required to add to the knowledge base and to provide additional material for a future meta review of psychoeducation initiatives specifically [5].

*4.2. Limitations*

This study has produced a set of tangible recommendations to improve the experience of waiting for a specialist ADHD assessment. Data were gathered and analysed by a student mental health nurse with limited experience of thematic analysis. They were, however, supervised by an experienced researcher to ensure fidelity to the cited six-stage process [19] was maintained. They were also independent of the service, ensuring objectivity. However, data were gathered from a small group of mainly male users in a single service, meaning the results may not be transferrable to other settings. That said, the views and suggestions of our participants had clinical face validity and were consistent with previous research which, together with the low risk of adverse effects of psychoeducation make these findings worthy of consideration.

**5. Conclusions**

The demand for specialist ADHD services in the UK is growing. Waiting for assessment, diagnosis, and treatment is at best frustrating and at worst detrimental to people's quality of life. Group psychoeducation packages have the potential to improve people's negative experiences of lengthy waits. These should cover a broad range of ADHD-related issues rather than having a narrow, diagnostic focus. Consideration should also be given

to additional one-to-one telephone support. Any such service developments should be evaluated to ascertain their impact and inform their continuous improvement.

**Author Contributions:** Conceptualization, J.P., F.O., B.G. and J.B.; methodology, J.P., B.G. and F.O.; validation, J.P., F.O., B.G. and J.B.; formal analysis, J.P., F.O. and B.G.; investigation, B.G. and F.O.; resources, B.G. and F.O.; data curation, B.G. and F.O.; writing—original draft preparation, B.G. and F.O.; writing—review and editing, B.G., F.O., J.P. and J.B.; supervision, J.P.; project administration, B.G. and F.O. All authors have read and agreed to the published version of the manuscript.

**Funding:** This research received no external funding.

**Institutional Review Board Statement:** The study was conducted in accordance with the Declaration of Helsinki, and approved by the Institutional Review Board (or Ethics Committee) of Sheffield Hallam University (using student research project approval process: UREC 3 24/11/21). It was also defined and approved by Sheffield Health and Social Care Trust as a service evaluation projects (06/12/21).

**Informed Consent Statement:** Informed consent was obtained from all subjects involved in the study.

**Data Availability Statement:** Not applicable.

**Acknowledgments:** The authors would like to thank the service users that participated in this study for their time and insights.

**Conflicts of Interest:** The authors declare no conflict of interest.

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
