# Peer review of "Service Users’ Perspectives on the Implementation of a Psychoeducation Group for People on the Waiting List of a Specialist ADHD Service: A Pilot Study"

_nursrep, doi:10.3390/nursrep13020058_

Round 1

Reviewer 1 Report

I thought this was a very nicely written manuscript which detailed the results of a focus-group investigation of services that should be provided to ADHD waitlist clients.  The methodology is clearly specified and the results which indicated 4 major themes were adequately reported and discussed.  Throughout the manuscript, the logical flow is perfect. Thus, I have only very minor suggestions:

In the Materials and Methods section, the authors state that their participant pool consisted of those with a diagnosis of ADHD, but in the results it appears that it consisted of both patients and family members (i.e., page 3 contains the statement “My son took medication, it caused him to have panic attacks and other things, he has had to come off it.”). Please clarify.

On page 2, the authors say “the coherence of each set of codes was considered and some statements were moved between themes until they were in the most appropriate place.” Please define “coherence” here for those of us who do not normally conduct focus-group research.

On page 4, there are a couple of sample quotes that mention filtering. Please give us a parenthetical (or other clarification) of exactly what is meant by a “filter.”

Once the above changes have been made, the paper should be published at the earliest convenience.  Nice work!

Author Response

Reviewer

Comment

Response

1

I thought this was a very nicely written manuscript which detailed the results of a focus-group investigation of services that should be provided to ADHD waitlist clients.  The methodology is clearly specified and the results which indicated 4 major themes were adequately reported and discussed.  Throughout the manuscript, the logical flow is perfect. Thus, I have only very minor suggestions:

Thank you for these encouraging words.

In the Materials and Methods section, the authors state that their participant pool consisted of those with a diagnosis of ADHD, but in the results it appears that it consisted of both patients and family members (i.e., page 3 contains the statement “My son took medication, it caused him to have panic attacks and other things, he has had to come off it.”). Please clarify.

This quote was from a service user with ADHD that happened to have a son with ADHD however we can see how its inclusion creates confusion.  We have therefore, in keeping with comment 9 (below) shortened the quote.

On page 2, the authors say “the coherence of each set of codes was considered and some statements were moved between themes until they were in the most appropriate place.” Please define “coherence” here for those of us who do not normally conduct focus-group research.

Term and explanation revised as requested.

On page 4, there are a couple of sample quotes that mention filtering. Please give us a parenthetical (or other clarification) of exactly what is meant by a “filter.”

Clarified that this is a colloquialism referring to the participant being unable to judiciously select which thoughts to verbalise.

Once the above changes have been made, the paper should be published at the earliest convenience.  Nice work!

Thank you

Reviewer 2 Report

1. In the data analysis section of the methods, it would be good if the actual process will be narrated to include a report of the figures or numbers of the statements, themes, etc.

2. Please clarify if participant 2 is a parent or a patient with ADHD

3. The way theme 2 is stated seems confusing.

4. The number of verbatims may be reduced. Choose what can best support your claim.

Author Response

Reviewer

Comment

Response

2

1. In the data analysis section of the methods, it would be good if the actual process will be narrated to include a report of the figures or numbers of the statements, themes, etc.

Coding tree included as fig 1

Please clarify if participant 2 is a parent or a patient with ADHD

This quote was from a service user with ADHD that happened to have a son with ADHD however we can see how its inclusion creates confusion.  We have therefore, in keeping with comment 9 (below) shortened the quote.

The way theme 2 is stated seems confusing.

Rephrased.

The number of verbatims may be reduced. Choose what can best support your claim.

We have reviewed these and reduced one quote to avoid confusion (see comments 2&7)

Reviewer 3 Report

I would like to thank the authors for the manuscript and for their interest in the subject. Alternative therapies such as the one you are investigating help and could improve the quality of life of people, and consequently improve their health.

Here are some possible improvements for your edition

Introduction: the introduction answers the question of why the work is being carried out and whether it contextualises current knowledge, but it could incorporate some additional information that helps us, such as previous work in the field or precision if there is any in the literature (another country).

Materials and methods.

although it is well explained that it is a qualitative research study, and they only present the discussion groups, I would like to know more and include more socio-demographic data, gender, ages, characteristics that help to see the focus group, number of sessions, and number of people, as well as the transcription programme, which they used apart from the zoom tool that they have already described.

Were the questions reviewed and validated by a group of experts?

results: you could incorporate a table, graph or figure to show clearly the main findings, as these are reliable evidence.

Here you have the described number of participants, (N=5) but in the previous section it would be good to put it in context.

Discussion:

I would make a revision to the discourse, incorporating some of the aforementioned apiortation.

Do you think it is important to suggest new lines of work?

Limit.

The analysis of the review by the student nurse in mental health, I think it is an important limitation to take into account, I would modify this paragraph, because of the ethical and professional interest of the nurse in the transcription of the data, without the accompaniment of an expert (I suggest modifying this part).

Author Response

Reviewer

Comment

Response

3

 would like to thank the authors for the manuscript and for their interest in the subject. Alternative therapies such as the one you are investigating help and could improve the quality of life of people, and consequently improve their health.

Here are some possible improvements for your edition

Introduction: the introduction answers the question of why the work is being carried out and whether it contextualises current knowledge, but it could incorporate some additional information that helps us, such as previous work in the field or precision if there is any in the literature (another country).

Thank you for confirming the relevance of this work.

Introduction expanded in line with comment.

Materials and methods.

although it is well explained that it is a qualitative research study, and they only present the discussion groups, I would like to know more and include more socio-demographic data, gender, ages, characteristics that help to see the focus group, number of sessions, and number of people, as well as the transcription programme, which they used apart from the zoom tool that they have already described.

We have clarified that this was a single focus group and that the zoom transcription was subsequently edited in Microsoft word.

The service is specified as being for working aged adults and we do provide the gender of participants.  Unfortunately, as with many qualitative studies, no additional demographic data were gathered.  Whilst we acknowledge the richness this may have offered; we hope you agree its omission does not detract from the merit of the article.  We have clarified that this was a single focus group and that the zoom transcription was subsequently edited in Microsoft word.

Were the questions reviewed and validated by a group of experts?

The questions were reviewed at the proposal stage by the service’s nurse consultant and the academic supervisor who is an NMC specialist practitioner.  We have clarified this in the text.

results: you could incorporate a table, graph or figure to show clearly the main findings, as these are reliable evidence.

Here you have the described number of participants, (N=5) but in the previous section it would be good to put it in context.

Coding tree now included as fig 1.

As per response to comment 11, unfortunately we are unable to provide any additional information about these participants.

Discussion:

I would make a revision to the discourse, incorporating some of the aforementioned apiortation.

Do you think it is important to suggest new lines of work?

We are mindful of the small scale of this project, and the fact that it is an evaluation of a specific service, rather than an attempt to generate generalisable findings.  We therefore feel more comfortable limiting our recommendations to ones that target this service specifically.  We have however, added a comment about the potential, in time to undertake some form of meta analysis of projects such as this.

Limit.

The analysis of the review by the student nurse in mental health, I think it is an important limitation to take into account, I would modify this paragraph, because of the ethical and professional interest of the nurse in the transcription of the data, without the accompaniment of an expert (I suggest modifying this part).

We have developed our comment about this in the limitations section to include the student’s supervisory arrangements. 

Author Response

Reviewer

Comment

Response

4

It is recommended to add a pilot study to the title, due to the sample size is small.

Added as suggested

Introduction The purpose of the research is clear and consistent. A few retouching is required for the waiting list of a specialist ADHD service. One is that the author should explain why it was a preliminary assessment for psychoeducation of ADHD service.

We have clarified the funding limitations that precluded anything other than a pilot study.

method

1. Could the author describe the preparation of holding the focus group's ability in this study?

2. Please explain why your method were used focus groups instead of individual in-depth interviews, because the sample number is five in your study?

3. Please describe clearly inclusion criteria about the participants and the demographic data.

Initial briefing now explained. 

Time and resource restrictions now included as the rationale for a single focus group.

Please see second paragraph of results section, plus response to comment 11 (above).

Result

4. It maybe typo. There are four themes in this study, but in line 99, “Three themes, each with several subthemes, emerged from the thematic analysis: 1) 99 support for psychoeducation in principle, ……. the research question is presented as theme 4) to help…

5. In addition to medicine for ADHD, please describe in detail issue about supportive psychoeducational interventions in theme 1.

6. Could the author explain more about Theme 3 (Suggested structure and approaches), because the participant said “Everyone’s different so what works for one person won’t work for another”

Due to the semi-structured nature of the focus group, participants provided feedback about the service that whilst not directly related to the proposed psychoeducation group, we felt should be captured and presented.  In the opening paragraph of the results section, we have clarified that, for convenience, these data are presented as a 4th theme. 

Thank you for highlighting the confusion caused by the way we had worded this section.  Our explanation of theme 1 has now been re-worded to clarify that responses were limited to psychoeducation as a augmentation strategy rather than any mention of other forms of intervention.

Thank you for drawing attention to this section which we have carefully reviewed to ensure it accurately reflects the point our participants raised.  They had preconceived concerns that group interventions may not meet the needs of the individuals.  However, in our discussion section we go on to explain that there is evidence to the contrary and that group work may actually have added benefits.

Discussion

7. Could the author describe more between the “Suggested structures and approaches” and “the need of person-centredness” ?

Discussion expanded re person-centredness, its underlying concept of empowerment, and evidence that this can be achieved through group psychoeducation.

Reviewer 5 Report

Thank you for the opportunity to comment on this study. I believe that the subject matter of the study is interesting and of interest to the target population. However, although I found the study interesting, I believe it is impossible to obtain a general overview. 

INTRODUCTION

Regarding the introductory section, there is a need for a greater theoretical and scientific foundation to support the theoretical justification, such as, for example, arguing under a theoretical basis what cognitive, metacognitive and dialectical behavioral therapy consist of.

More specifically, there is empirical evidence to suggest psychoeducation groups are a feasible [10] and acceptable [11]way to increase adults' knowledge of ADHD [12] and ability to manage their symptoms [13]. (WHAT EMPIRICAL EVIDENCE? JUSTIFY IT?)

It is recommended that the citations in which they rely on the scientific literature, although some are current, most are below 2010, so I invite the authors to inquire about more current literature on the subject of study that they develop. 

2. Materials and Methods

In this section, I would place "Data collection" and "Data analysis" as subsections.

Regarding data analysis and results, I consider it to be quite limited. 

What kind of statistical software did you use?

It would give a scientific value to your article if you employed an empirical-analytical and descriptive methodology and made use of content analysis techniques that NVivo software, for example, would provide you with. In addition, it would be good to perform mixed and contextual analyses in relation to specific keywords of your study. So I recommend to the authors that, the statistical section should be improved so that their research increases in scientific quality. 

In the discussion, authors should explain their findings in more depth. The authors should broaden the implication of this research.

Limitations are not strongly discussed and the authors should include a stronger call for future research.

Many of the bibliographic references do not conform to the style required by the journal, so they should adapt them correctly.

I encourage the authors to make these improvements, which will surely make their study more scientifically sound.

Author Response

5

INTRODUCTION

Regarding the introductory section, there is a need for a greater theoretical and scientific foundation to support the theoretical justification, such as, for example, arguing under a theoretical basis what cognitive, metacognitive and dialectical behavioral therapy consist of.

More specifically, there is empirical evidence to suggest psychoeducation groups are a feasible [10] and acceptable [11]way to increase adults' knowledge of ADHD [12] and ability to manage their symptoms [13]. (WHAT EMPIRICAL EVIDENCE? JUSTIFY IT?)

It is recommended that the citations in which they rely on the scientific literature, although some are current, most are below 2010, so I invite the authors to inquire about more current literature on the subject of study that they develop.

Introductory section developed and more contemporary references added.

5

 Materials and Methods

In this section, I would place "Data collection" and "Data analysis" as subsections.

Sub-sections added

5

What kind of statistical software did you use?

No statistical software was used to ensure the novice researcher remained “close” to the data as advocated by Green & Throgood (2018, p281), truly understanding the nature of the task rather than becoming prematurely reliant on software.

5

It would give a scientific value to your article if you employed an empirical-analytical and descriptive methodology and made use of content analysis techniques that NVivo software, for example, would provide you with. In addition, it would be good to perform mixed and contextual analyses in relation to specific keywords of your study. So I recommend to the authors that, the statistical section should be improved so that their research increases in scientific quality.

We agree that different forms of qualitative data analysis exist and can each offer different benefits.  However, we selected thematic analysis as a flexible, straightforward and widely used, method that has merit in its own right (Braun & Clarke, 2006).  We respectfully believe that, with the revisions made in response to reviewer comments, this paper has scientific value in its current guise.

5

In the discussion, authors should explain their findings in more depth. The authors should broaden the implication of this research.

This project was granted ethical approval on the premise that it was a service evaluation.  The HRA decision tool: https://www.hra-decisiontools.org.uk/research/ precludes such projects offering generalisable findings.  We have therefore made some modest revisions to the discussion section (in line with other suggestions) but do not feel we can legitimately make any wider claims.  We trust this is satisfactory. 

5

Limitations are not strongly discussed and the authors should include a stronger call for future research

Limitations have been expanded in response to comment 15 (above) and we have however, added a comment about the potential, in time, to undertake some form of meta-analysis of service-specific projects such as this.

5

Many of the bibliographic references do not conform to the style required by the journal, so they should adapt them correctly.

Thank you for prompting us to check this.  The journal’s authoring guidelines state that any referencing format is acceptable however we have reviewed the reference list for consistency.

Round 2

Reviewer 5 Report

Dear authors,

I have reviewed the article and I thank you for taking into consideration the recommendations for improvement that I indicated. The article has improved considerably, so I recommend that the article be published in its form.

Finally congratulations to the authors for the work done.

Best regards,

Jose Manuel.